# South China Sea documents the transition from wide continental rift to continental break up

Hongdan Deng [1,2✉], Jianye Ren[2,3], Xiong Pang[4], Patrice F. Rey[5], Ken R. McClay[6], Ian M. Watkinson[7], Jingyun Zheng[4] & Pan Luo[2]

During extension, the continental lithosphere thins and breaks up, forming either wide or narrow rifts depending on the thermo-mechanical state of the extending lithosphere. Wide continental rifts, which can reach 1,000 km across, have been extensively studied in the North American Cordillera and in the Aegean domain. Yet, the evolutionary process from wide continental rift to continental breakup remains enigmatic due to the lack of seismically resolvable data on the distal passive margin and an absence of onshore natural exposures. Here, we show that Eocene extension across the northern margin of the South China Sea records the transition between a wide continental rift and highly extended (<15 km) continental margin. On the basis of high-resolution seismic data, we document the presence of dome structures, a corrugated and grooved detachment fault, and subdetachment deformation involving crustal-scale nappe folds and magmatic intrusions, which are coeval with supradetachment basins. The thermal and mechanical weakening of this broad continental domain allowed for the formation of metamorphic core complexes, boudinage of the upper crust and exhumation of middle/lower crust through detachment faulting. The structural architecture of the northern South China Sea continental margin is strikingly similar to the broad continental rifts in the North American Cordillera and in the Aegean domain, and reflects the transition from wide rift to continental breakup.

[1] Hubei Key Laboratory of Marine Geological Resources, China University of Geosciences, 430074 Wuhan, China. [2] College of Marine Science and Technology, China University of Geosciences, 430074 Wuhan, China. [3] Key laboratory of Tectonics and Petroleum Resources of Ministry of Education, China University of Geosciences, Wuhan, China. [4] CNOOC Ltd. Shenzhen Branch, 518054 Shenzhen, China. [5] Earthbyte Research Group, Basin Genesis Hub, School of Geosciences, The University of Sydney, NSW 2006 Sydney, Australia. [6] Australian School of Petroleum, Adelaide University, North Terrace, SA 5000 Adelaide, Australia. [7] SE Asia Research Group, Department of Earth Sciences, Royal Holloway University of London, Egham, Surrey TW20 0EX, UK. ✉email: denghongdan@gmail.com

Continental extension typically results in one of two dominant modes of rifting: 'narrow rift' and 'wide rift' modes[1–3]. In narrow rifts, extension and thinning is concentrated in regions no larger than the thickness of the continental lithosphere. In contrast, in wide continental rifts exemplified by the ongoing extension of the North American Cordillera and the Aegean domain, continental thinning occurs over a continental width many times the thickness of the lithosphere[1]. This mode involves intense ductile flow of the lower crust and upper crustal boudinage, facilitated in many cases by the formation of detachment faults[4–9]. As a part of the wide continental rifts, metamorphic core complexes (MCCs) are crustal-scale domal structures flanked by low-angle detachment faults that tectonically juxtapose upper crustal, brittlely deformed rocks with lower crustal, ductilely deformed metamorphic rocks[5,10–12]. The massive ductile deformation is quite different to that observed at cold magma-poor margins, such as the West Iberia-Newfoundland margins, where progressive cooling and embrittlement of the lower crust dominates extension that leads to mantle serpentinization and exhumation at the ocean–continent transition zone[13–17]. In contrast, the ocean–continent transition in hot magma-poor continental margins, such as the Woodlark Basin, exhibit more predominance of ductile extension and magmatism in the lower crust with no mantle exhumation[18–21]. While research on wide continental rifting has focused on collapsed orogenic crust[5,6] and studies on highly extended crusts have focused on wide continent to ocean transitions[16,22], few studies have considered the possible link between the two[21]. Furthermore, despite the prediction that continuous wide rifting of continental lithosphere could ultimately lead to passive margin formation[23,24], well-defined and characteristic MCCs have not been identified in situ on highly extended crust, and therefore the transition from wide continental rifting to breakup has not been completely understood. This is partly hampered by limited geophysical data coverage and resolution, and partly due to the imaging problems prevalent in two-dimensional seismic time sections in submarine areas of the distal continental margins.

Surrounded by the Pacific and Indian oceans and the Eurasian plates (Fig. 1a), the South China Sea is the largest marginal sea of the western Pacific[25]. Numerous studies have shown that southeastern China recorded a long history of northward subduction of the proto-South China Sea starting in the Triassic and ending during the latest Cretaceous, involving dominant back-arc extension[26–28]. Continued and significant continental extension of this domain started in the Early Eocene (~52 Ma)[29], characterised by punctuated and diachronous phases of extension leading to final continental breakup in the Early Oligocene (~30 Ma)[30,31]. Extension and rifting resulted in an ultra-wide (up to 1000 km) northern South China Sea passive margin (Fig. 1 and Supplementary Fig. 1) and its southern counterpart (up to 500 km)[7,32–34], fringed by a stripe of highly extended (<15 km) continental ribbons tapering into the oceanic lithosphere[35–37].

Using high-resolution seismic data on the highly extended northern continental margin of the South China Sea (Fig. 1a), we have clearly imaged the marginal crust down to 8–12 s two-way-time (25–35 km) on regional two-dimensional seismic lines (Fig. 1b, c and Supplementary Fig. 2) and 10 km in depth-migrated three-dimensional seismic coverage, the latter at a spatial resolution of a few ten of metres. We then compare our seismic studies with well-studied exposed analogues from the North American Cordillera, the Aegean Sea, and the Woodlark Basin. Our analysis allows us to critically evaluate the involvement of MCCs and to propose that margins of the South China Sea are typical of rifting of thermally weakened active margins.

## Results

**Subdetachment deformation.** Along a north–south regional seismic profile (X–X′ on Fig. 1b, c and Supplementary Fig. 2), the northern continental margin of the South China Sea exhibits strong upper crustal boudinage. Indeed, from north to south, the thickness of the upper crust changes from ~5 s two-way travel time (TWT) (13–16 km), along the intrabasinal highs of the proximal domain, to 0–1 s TWT (<~3 km) below the Liwan subbasin, and to 4 s TWT (~13 km) in the distal domain further south (Fig. 1b, c). An isocline of seismic energy is organised into a pattern documenting 30–40 km long nappe folds (Fig. 1b, c) below, and sub-parallel to, the Liwan detachment fault (Fig. 2). The geometry of these deformations is interpreted based on the distinctive patterns of the high- and low-amplitude reflectors (Fig. 3). The envelope of the low-amplitude homogeneous reflection shows a concentric (or diapir-like) pattern in the core, mantled by layered high-amplitude reflectors (Fig. 3a). The axes of the concentric and mantling reflectors are tilted and have a small angle against the upper detachment fault. Both the high- and low-amplitude reflections show continuous axes from the homogeneous core to the layered mantle, and both the diapir-like structure and the isoclines verge to the south, in agreement with the inferred shear sense along the Liwan detachment fault (Fig. 3). The sub-domes (D1, D2, and D3) are delineated by the coherence and azimuth attributes of the detachment (Fig. 2b, c), which is less smooth and lacks obvious grooves compared to the intact surface. These domes generally trend E–W and have shorter axes than that of the N–S corrugations. Grooves and corrugations have been deformed by the sub-domes, as indicated by separation of otherwise continuous striations and fold axes (Fig. 2a). In the cross-sections, the three domes D1, D2, and D3 correspond to the arching of subdetachment strata and faults (Fig. 3), implying that some of the exhumed material from the deep crust impinged into the supradetachment basin.

**Geometry of detachment fault.** The Liwan detachment fault has been deformed (Figs. 1b, c–3) and shows fold pairs or corrugations, domes, grooves, and supradetachment faults (Fig. 2a–c). The orientation of the detachment surface is locally variable, but generally dips about 6–10° to the south (Fig. 3). On the detachment surface, the corrugations are parallel to the grooves that trend 005° in the north and slightly curve to 010° in the south (Fig. 2d–g and Supplementary Data 1). The detachment fault corrugations have crest-to-trough amplitudes of 2–3 km and crest-to-crest wavelengths of 10–15 km, extending >35 km along fault dip and plunging to the south. The grooves are remarkably well developed and are parallel to each other, with some groove lengths up to 20 km (Fig. 2a–c). The supradetachment faults trend E–W in the north and south and trend 065° in the central part of the detachment (Fig. 2e–g and Supplementary Data 2). These faults offset the grooves and corrugations (Fig. 2a–d) and suggest that they developed in the later stage of detachment fault formation.

**Supradetachment basin.** Stratigraphic architecture of the supradetachment basin shows progressive southward stacking of the Eocene–Oligocene (Tg–T60) sediments (Fig. 3b), accommodated by depositional space generated by top-to-the-south unidirectional removal of the hangingwall block in the Eocene. Over the southern end of the detachment fault, Oligocene (T70) stratigraphic onlaps (green arrows in Fig. 3) indicate progressively vertical basin subsidence of 2–3 km. Underneath the Oligocene basin, the Lower Eocene (Tg–T80) units are missing, and the Upper Eocene (T80–T70) units are more attenuated than on the flanks (Fig. 3). These observations suggest pure-shear extension

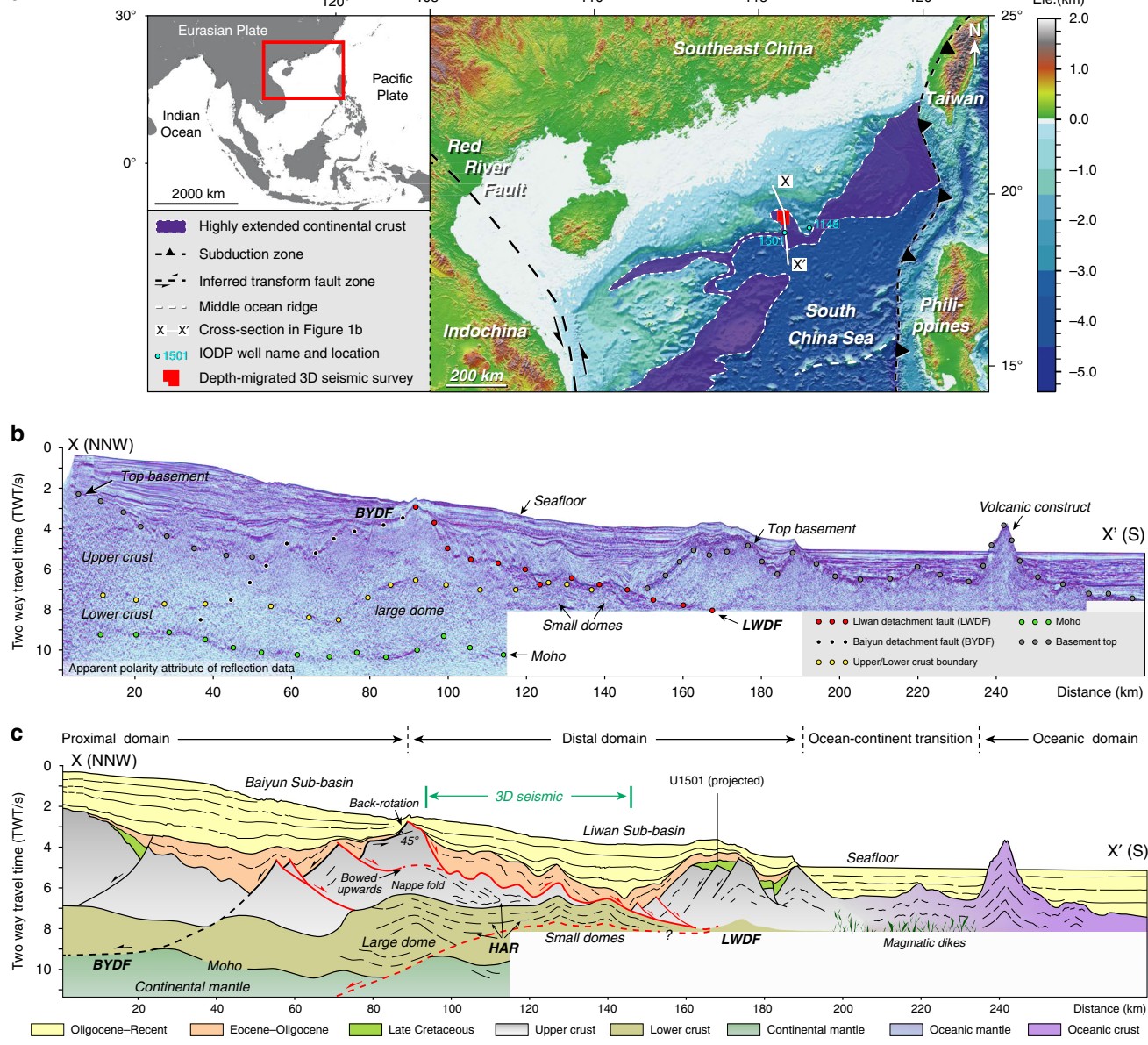

**Fig. 1 Location and cross-section of the northern South China Sea continental margin. a** Highly extended continental margin of the northern South China Sea. The extent of highly extended continental margin is modified from previous research[35]. **b, c** Seismic interpretation of X–X' (white line in **a**) showing dome structures indicated by high-amplitude reflectors and detachment faults flanking the large dome between Baiyun and Liwan sub-basins. The upper part of the dome structures consists of a nappe fold, across which the crustal thickness thins quickly from 7 s in the north to 3 s in the south. BYDF Baiyun detachment fault, LWDF Liwan detachment fault, HAR high-amplitude reflector.

during the Oligocene that localised at the southern end of the Liwan detachment fault. The timing of the localised extension is coeval with the upward doming of D1, D2, and D3 that are topped by the Oligocene unconformity (T70 in Fig. 3).

## Discussion

The North American Cordillera and the Aegean Sea are regions that exemplify the wide rift mode of continental extension[1,2,5]. In both regions, hangingwall supradetachment basins, detachment faults, and footwall subdetachment nappes and sub-domes constitute the system of MCCs that are well-exposed and have been extensively studied in the last four decades[5,38–43]. The Woodlark Basin is a region of continental extension in a relatively thick crust (>26 km) that exhibits diapiric exhumation of lower crust (MCCs)[18,19]. In all these extended back-arc regions the upper plate has been thermally weakened by subduction related

processes. However, unlike the South China Sea, ongoing extension has not yet reached the transition from wide rifting to extreme thinning and to drifting. Using high-resolution two- and three-dimensional seismic data (Figs. 1–3) along the northern margin of the South China Sea, we have identified an association of structures that strikingly resemble those documented in the North American Cordillera and Aegean domains as well as the Woodlark Basin and analogue and numerical models.

The Liwan detachment fault shows characteristic corrugations, domes, and grooves (Fig. 2) that are commonly observed in, and are genetically linked to, the MCC systems (Figs. 4 and 5)[41,44–47]. The aspect ratio (length/width) of the corrugations and domes of the Liwan detachment fault is about 2.5, which lies within the range of 2–3 for most gneiss domes of MCCs[48] (Fig. 4a and Supplementary Data 3). The corrugations of the Liwan detachment fault are different from those of the shallow-dipping (<10°) detachment

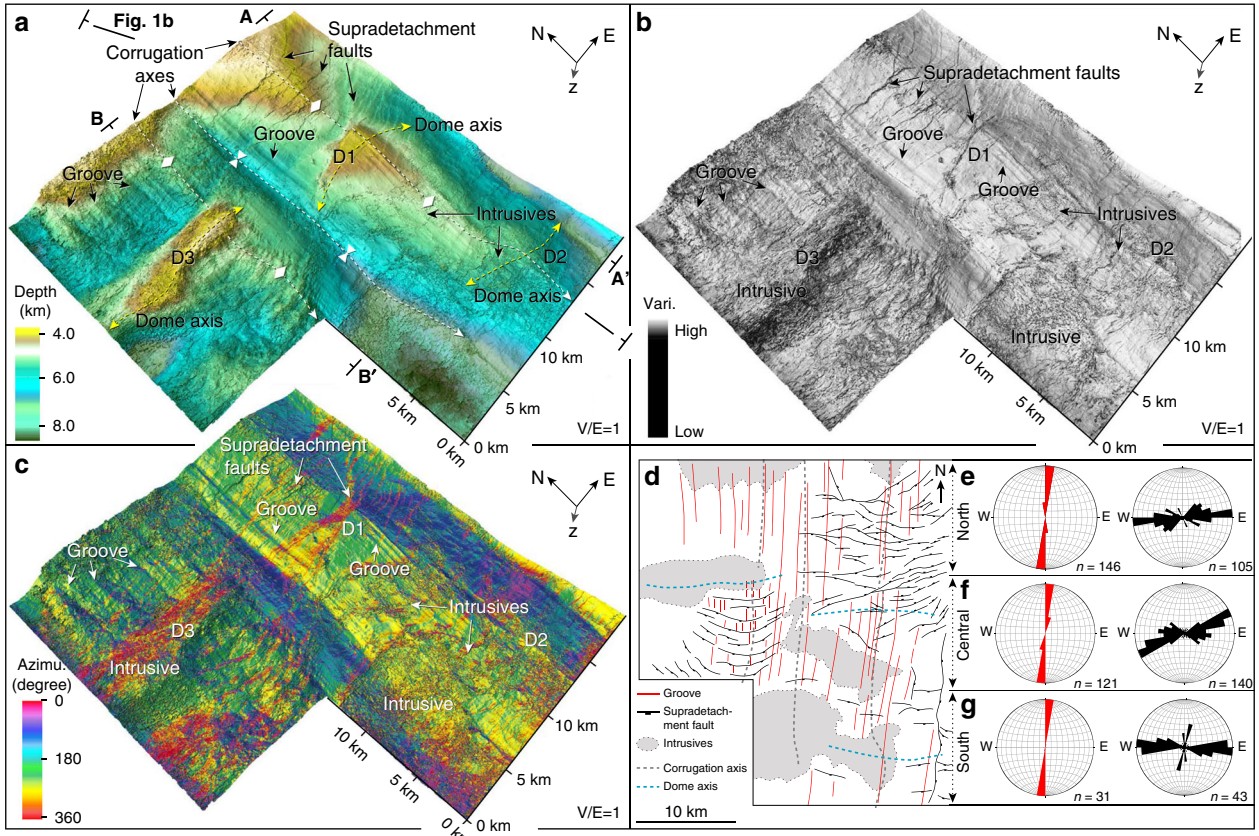

**Fig. 2 3D perspective views of depth, attribute, and azimuth attributes of the Liwan detachment fault and the interpretation and orientation of grooves and secondary faults on the fault plane. a–c** Depth, coherence, and azimuth attributes extracted along the Liwan detachment fault. Note that the attribute maps show grooves, supradetachment faults, corrugations, intrusives, and sub-domes on the fault plane. The corrugations mostly trend N–S, parallel to the grooves. The sub-domes are numbered as D1, D2, and D3 and are E–W-trending. The A–A′ and B–B′ are cross-sections shown in Fig. 3. **d** Interpretation of the grooves, supradetachment faults, and intrusives on the Liwan fault plane. **e–g** Orientation plots of grooves (red) and supradetachment faults (black) from the north, central, and south. The plots show that the grooves generally trend 005° but some segments slightly change by 010° to the central and the south. The supradetachment faults mostly trend 085° in the north, 060° in the central, and 090° in the south. The grooves and faults are sampled at every square kilometre and their orientation data are listed in Supplementary Data 1 and 2.

of the West Iberian margin[49,50]; the latter have smaller dimensions (length < 8.0 km, width < 1.1 km) and larger aspect ratio (6–16) (Fig. 4b and Supplementary Data 3). The differences in size and in aspect ratio of hot and cold continental margins may suggest that temperature plays a significant role in shaping the geometry of MCC corrugation and doming of detachment faults.

The formation of an antiformal–synformal geometry of the corrugated detachment fault (Figs. 2 and 5a) is possibly associated with a uniaxial stress field that yields compression perpendicular to the hangingwall transport direction[46]. Similar to the corrugations, grooves on the detachment surface are a fundamental part of the fault geometry (Fig. 2), which developed analogous to the fault surface striations along detachment faults on the continent[45] and exposed on the seafloor in oceanic core complexes[51]. The E–W domes (D2 and D3 in Figs. 2a–c and 3) represent active exhumation of lower crust in the Early Oligocene. These domes' axes are perpendicular to the extension and are analogous to extension-perpendicular domes of the Evvia-Mykonos MCCs in the Aegean domain[41] and D'Entrecasteaux-Dayman-Suckling MCCs in the Woodlark Basin, Papua New Guinea[18,19,52]. The presence of lower crustal doming favours strong deep crustal ascent and abnormally hot conditions underneath the detachment fault which becomes weakened and partially molten, enhancing strain localisation[3,7,53,54]. The Liwan detachment fault shows back-rotation of the footwall block bounded by southward dipping faults that exhibit a bow-upward geometry at depth

(Fig. 1b, c). The concave-upwards geometry of the detachment fault was likely locked up as extension and exhumation continued and was later replaced by the formation of new detachment fault in the hangingwall[55,56].

Subdetachment structures include nappe folds, intrusions, and domes (Fig. 1b, c), among which the large dome exhibits a symmetrical and upright geometry (Supplementary Fig. 3) that is mostly observed in migmatite-cored MCCs[12,57]. In the footwall of the southern flanking detachment, the highly laminated middle/lower crust reflectors present an isoclinal geometry (Figs. 1b, c and 3) evocative of the pattern observed in nappe folds[58]. These high-amplitude reflections may represent mylonite foliations wrapped around the rising domes that have been transposed by top-to-the-south shear imposed by the hangingwall movement[58,59]. The low-amplitude homogeneous reflection below the Liwan detachment fault shows concentric zonation of reflectors (Fig. 3a) with steps that pinch out to the south (Fig. 3b). The steps are commonly observed in igneous intrusions with characteristic magma fingers[43,60]. A similar magmatic origin also applies to the concentric zonation of low-amplitude reflections in the core that progressively evolve into high-amplitude reflections in the mantle (Fig. 3a). This can be explained by a facies change from coarse-grained, foliated granodiorite in the margin to fine-grained, nonfoliated granodiorite in the core that is analogues to plutonic intrusions in Serifos, Naxos, and Ikaria[43,61–63]. The presence of volcanic material has been inferred previously by

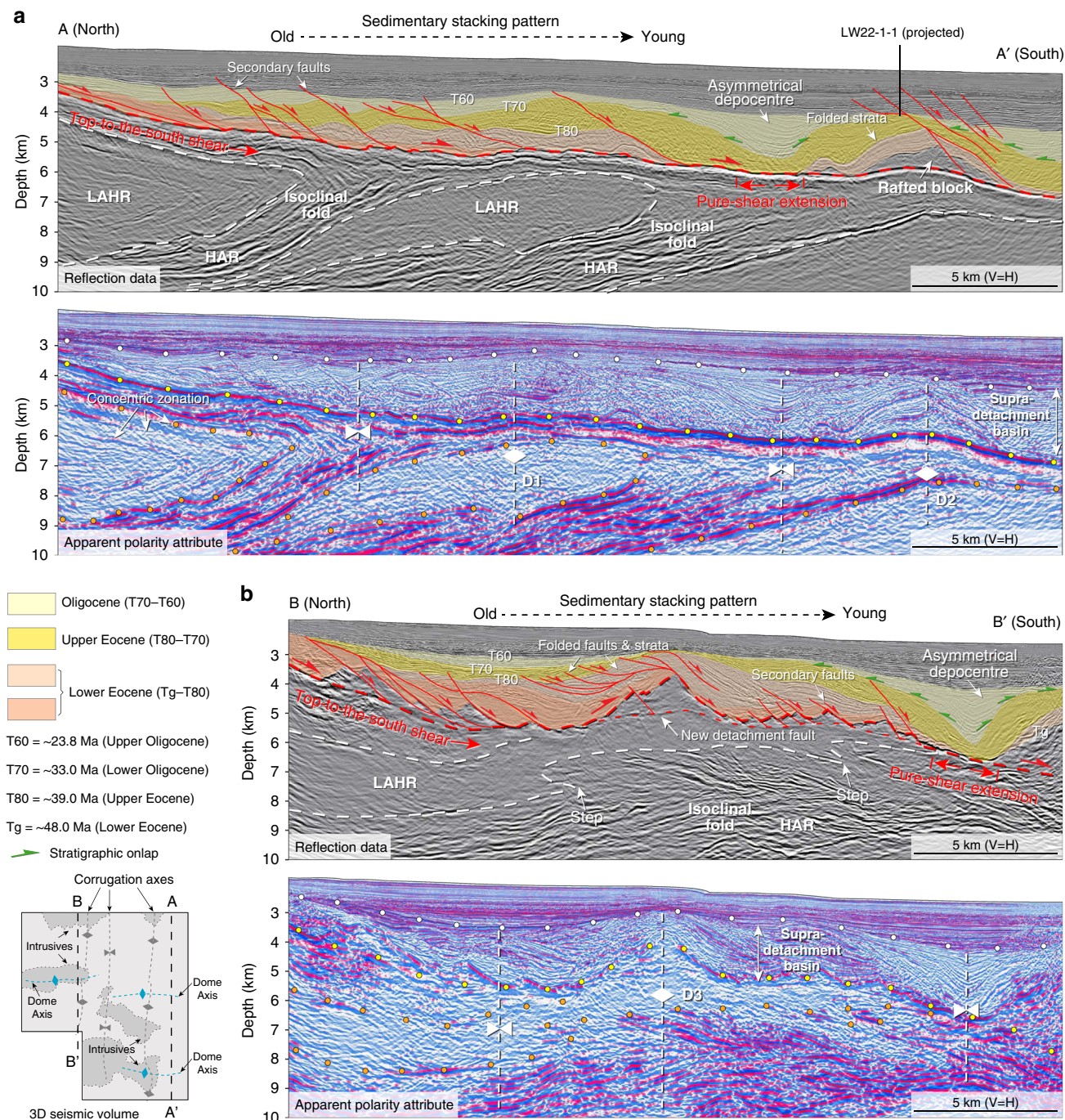

**Fig. 3 Interpretation of reflection data and apparent polarity attribute parallel to the dip of the Liwan detachment fault. a** Seismic section on the eastern side of the study area; **b** seismic section on the western side of the study area. Note that both seismic sections show the detachment fault geometry, stratigraphic architecture of the supradetachment basin, and subdetachment deformation in the southern flank of the large dome in Fig. 1b. See Fig. 2 for sections location and the sub-domes (D1, D2, D3). The white circles denote the upper boundary of the supradetachment basin. The yellow circles delineate reflections from the Liwan detachment fault. The brown circles mark the boundary of low-amplitude homogeneous and layered high-amplitude reflections. The white vertical dashed lines represent fold axes. The southeastward younging of Eocene sediments suggest top-to-the-south shear associated with detachment faulting. The asymmetrical depocentre is defined by the onlaps (green arrows) on the flanks. It may indicate pure-shear extension in the south of the detachment fault, which occurred simultaneously with the upwelling of D1, D2, and D3 and developed between T70 and T60 (Oligocene). LAHR low-amplitude homogeneous reflection, HAR high-amplitude reflection.

wide-angle seismic data[36]. However, the viscosity of the intrusive body may not be very high, since it shows no evidence of piercing into the overlying sedimentary succession of the supradetachment basin; instead, it arches the basin (Fig. 3).

The development of a hangingwall supradetachment basin is dominated by southward younging of laterally stacked sediments followed by localised vertical accumulation of asymmetrical subsidence in the southern end (Fig. 3), which as a whole were southward-tilted by the uprising dome below the Liwan detachment fault. Such asymmetrical basins are best observed elsewhere in salt provinces[64] but the distal continental margin of the South China Sea shows no evidence of thick evaporite

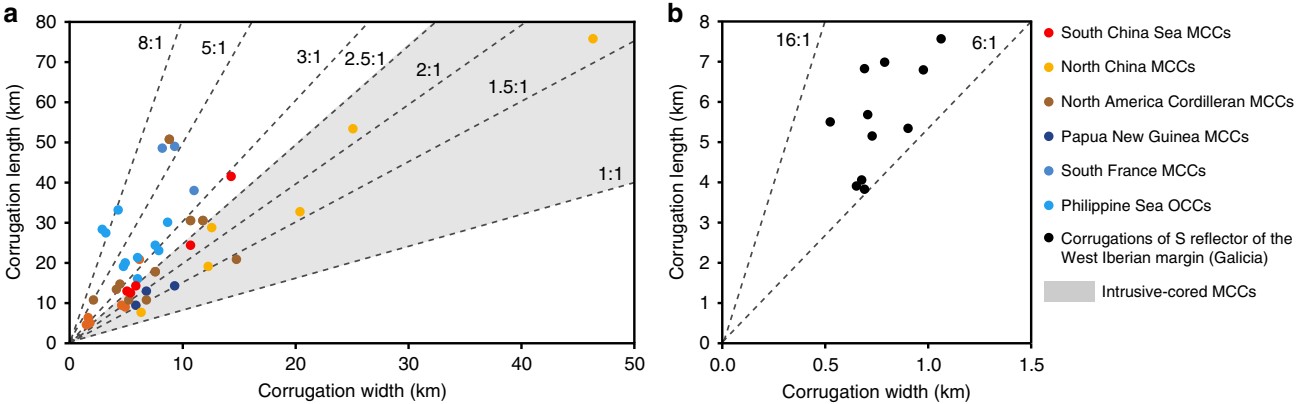

**Fig. 4 Dimensions and aspect ratios of domes of metamorphic core complexes (MCCs).** (**a**) and corrugations of the S reflector (**b**). Note that intrusive-cored domes have an aspect ratio of 1–2.5. Aspect ratios of oceanic core complexes are 2.5-8 or larger, similar to that of corrugations of the S reflector. The aspect ratios of the corrugations and domes of the northern South China Sea are 2–3. Corrugations of the S reflector on the Galicia margin generally have smaller length and width than those of metamorphic core complexes and oceanic core complexes (OCCs). The data sources and the measurements are listed in Supplementary Data 3.

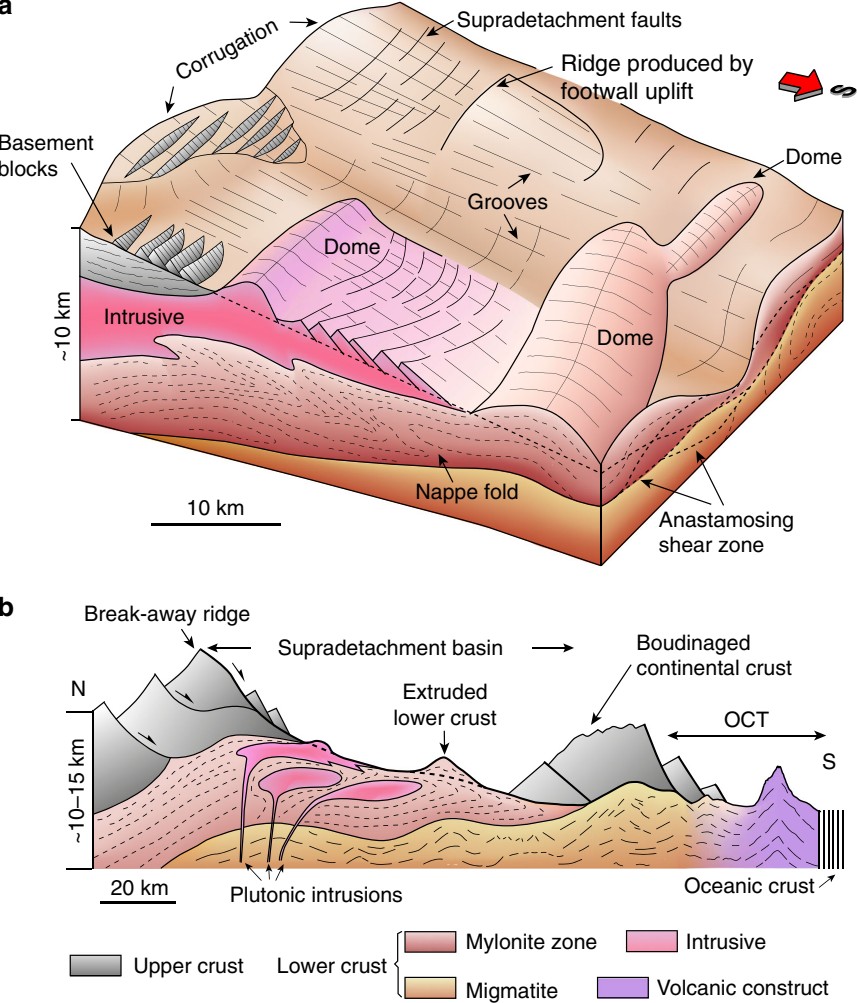

**Fig. 5 Metamorphic core complexes (MCCs) of the northern South China Sea showing characteristic folded detachment fault, crustal-scale nappe folds, migmatite- and/or pluton-cored domes. a** Three-dimensional block diagram illustrating the detachment surface and the footwall deformation associated with the faulting process. Note that the doming of the detachment includes N–S and E–W corrugations. Intrusives and nappe folds verge to the south along the southward shearing of the detachment. **b** Conceptual sketch cross-section parallel to the extension direction showing the highly extended continental crust and the ocean–continent transition. Upper crust boudinage in the highly extended continental margin evolves down-dip into the ocean–continent transition (OCT) and oceanic lithosphere.

deposition[29,32,37,65,66]. In the regional context of the South China Sea, their timing of localised basin formation corresponds to the final stage of the continental breakup[67,68]. Hence, the formation of asymmetrical subsidence at the distal part of the hyper-extended margin is possibly the result of an accelerating extensional rate and increasing geothermal gradient that was accompanied by reduction of lower-crustal viscosity and strain localisation during continental breakup[5,54]. Therefore, the supradetachment basin was controlled by the development of the Liwan detachment fault that was dominated by simple-shear and locally by pure-shear extension.

Combining the analysis of depth-migrated, high-resolution three-dimensional seismic data interpretation and well-known natural analogues, we have documented along a highly extended (<15 km) marginal crust an association of structures typical of wide rifts (Fig. 5): MCCs with supradetachment basins, a corrugated and grooved detachment fault, and subdetachment deformations on the continental margin of the northern South China Sea (Figs. 1–3). Specifically, we document that (i) formation of extensional-parallel and extensional-perpendicular folds or domes and grooves are genetically linked to the formation of MCCs (Fig. 5a); (ii) deep crustal domes associated with vertical ascent (upright domes) and lateral flow (nappe folds) of partially molten mass suggest intense ductile deformation and middle/lower crust exhumation in the distal domain; (iii) the Liwan detachment fault was chiefly controlled by simple-shear deformation during the Eocene extension and by localised pure-shear deformation during the Oligocene extension, the latter coeval with deep crust extrusion and strain localisation during continental breakup.

These structures (Fig. 5) exhibit striking similarities to the MCCs of the North American Cordillera, the Aegean Sea, and the Woodlark Basin[18,19], implying that the lower crust of the South China Sea distal margin was rather hot and weak during the development of continental thinning prior to and during the onset opening of the South China Sea. This context explains large-scale flow of crustal material associated with extensional instabilities due to a thermal anomaly and the formation of large detachment fault systems to the north of the Liwan sub-basin that significantly thinned the crust. Highly extended hot continental margin exemplified by this study is dominated by distributed upper crust necking and boudinages coupled with exhumation of middle/lower crustal (Fig. 5b), in stark contrast to large scale mantle exhumation and usually one necking domain of cold continental margins[13,15,16,22,69]. The northern margin of the South China Sea documents the development from a wide continental rift to an highly extended continental margin and to breakup in the context of a weakened lithosphere, and therefore provides crucial insights into the transition from an advanced wide continental rift (e.g. North American Cordillera/Aegean Sea) to continental breakup.

## Methods

**Three-dimensional seismic data and bathymetry data**. The three-dimensional (3D) seismic dataset was acquired by CNOOC in 2011 using two airgun arrays spaced 50 m apart, fired every 25 m at 2000 psi. The total volume used was 7750 in ref. [3], towing at a depth of 6 ± 0.5 m throughout the survey. The data were collected using twelve hydrophone streamers, each 600 m in length, containing 480 channels, regularly spaced at 12.5 m. The data were recorded for 8192 ms at a 1-ms sample rate. The data were reprocessed and generated pre-stack depth migration data by PGS in 2012. It has an inline (N–S) and crossline (E–W) spacing of 12.5 m × 12.5 m, 5 m vertical sample rate, covers an area of ~1500 km$^2$ and has a record length of 10 km. There are 3142 inlines and 3816 crosslines in the data volume. The main frequency of the reprocessed volume is 30–45 Hz. The velocity model applied for the depth conversion of the seismic volume is consistent with the velocity from ocean bottom seismic survey OBS93 (ref. [36]).

The bathymetry map of the northern South China Sea and adjacent areas was made with the Generic Mapping Tool programme version 6 (ref. [70]). The data for

plotting this map are downloaded from GEBCO one minute grid (https://www.gebco.net/data_and_products/gridded_bathymetry_data/).

**Detachment fault mapping and visualisation**. Mapping of the detachment fault was performed using GeoFrame v2012 on the Pre-stacked depth seismic volume. The detachment surface was interpreted at 50 × 50 grid spacing (600 m × 600 m) and was then autotracked to map the entire volume based on the seeded grids. Refined interpretation of distorted detachment surface due to the presence of intrusion was performed with 10 × 10 grid, which were autotracked for a second time. The resultant horizon was not smoothed or filtered in order that the subtle deformation could be maintained.

3D perspective of the detachment fault was manipulated in the Geoviz module of the GeoFrame. The detachment surface was viewed from a bird perspective looking down towards the northwest and a light source was positioned in the northwest at ~45° above the display.

**Geometric and amplitude attributes**. Two geometric attributes, azimuth and variance, were used to highlight subtle shape changes and to validate the depth structure of the mapped detachment fault. The azimuth attribute calculates the azimuth at each sampling point (3 × 3 sample rate). In the colour-coded azimuth attribute, the E–W faults were displayed in red and the N–S lineation were displayed in blue and yellow. The variance attribute is derived from measuring of discontinuities of the reflection volume in the depth window ±5 m around the mapped horizon at 3 × 3 sampling rate. High discontinuities such as faults and intrusive induced distortions is highlighted in black, whereas continuous reflection is marked as marked as white.

Apparent polarity attribute displays local maxima of Hilbert envelope. It measures both reflection strength and polarities of the seismogram. Default parameters were used for the seismic attribute extraction.

**Measuring grooves and secondary faults**. Orientation of the grooves and secondary faults of the detachment surface was manually measured. The detachment fault with a dimension of 32 km wide and 35 km long in map-view were divided into 1 km × 1 km grid. The measured results were plotted with Stereonet version 10.4.2 publicly provided by Richard W. Allmendinger.

## Data availability

The authors declare that the proprietary 2D and 3D seismic data used in this paper are from CNOOC. Other data are accessible within this article and its supplementary files.

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

## Acknowledgements

H.D. and J.R. were funded by National Natural Science Foundation of China (Grants No. 41830537 and No. 41902123). P.F.R. acknowledges support from ARC-IH130200012. The authors acknowledge CNOOC for allowing them to publish these data.

## Author contributions

H.D. devised the project and conducted the interpretation and analyses, and wrote the original draft; J.R. lead the funding acquisition, revised part of the manuscript, and jointly discussed the interpretation and results with X.P. and H.D.; X.P. edited part of the revised manuscript; P.F.R. and K.R.M. wrote part of the draft, improved seismic data imaging and writing, and provided information on analogues comparison; I.M.W. edited the manuscript and suggested for further improvement of the figures and writing. P.F.R., K.R.M., and I.M.W. participated in all stages of paper revisions. J.Z. and P.L. supported the seismic data interpretation.

## Competing interests

The authors declare no competing interests.
