## [Peer Review File · Nature Communications]

REVIEWER COMMENTS

Reviewer #1 (Remarks to the Author):

This is a very important paper. Its topic is very timely and the data set is first-class. Distributed extension with ductile deformation, metamorphic core complexes (MCCs) and detachments have been described in many regions with hot and weak crust and lithosphere, such as the Basin and Range and the Mediterranean region, among others. On the other hand, passive margins are often associated with very different extension mechanisms and rheologies, much colder and resistant. Recent offshore investigation with high-resolution seismic profiles have shown that many passive margins show much more ductile deformation during rifting and association with magmatism, hence the concepts of hot and cold margins developed in several recent papers. The ductile deformation and low-angle normal faults seen on these hot margins are partly similar to those observed in regions with detachments and MCCs, but the resolution of these nice images are still far from the field observations in the Basin and Range or the Aegean where detailed cross-sections have been described showing the relations between ductile deformation, high-temperature metamorphic domes, syn-extension intrusions, detachments and syn-extension basins. This paper fills this gap in a magnificent way and definitely makes the connection between wide-rift dynamics and continental break-up. The seismic images shown here have an unprecedented resolution and they show a striking similarity with the field observations made by geologists in the Basin and Range or the Aegean. The most striking features are the detailed geometry of the Liwan detachment, its corrugations and its relations to sedimentation above and the asymmetry of structures below the detachment, calling for simple shear deformation compatible with the kinematics of the detachment. The nappe folds and the geometry of intrusions are also very similar to field observations in the Aegean and they show that the entire crust was sheared in a consistent way during rifting. This is the first time I see seismic data across a series of MCCs at crustal scale with such a definition that only offshore studies can provide. This paper shows (1) that Aegean-like extension can lead to continental breakup and that hot margins are more frequent than expected and (2) that the crustal-scale deformation features proposed by geologists in the Basin and Range or the Aegean are real. It also definitely shows that the South China Sea region was supported by a hot and weak lithosphere during the Eocene and Oligocene. This paper should then be published as soon as possible so that these magnificent images are made available to the scientific community. This paper would likely attract a lot of citations.

I then list some issues, some are very minor, others more serious, which should however be discussed or corrected:

1- Abstract: how do you define hyper-extension? This is ambiguous in the literature and it is often associated with mantle exhumation in the OCT. Clearly this is not the case here and recent drilling in the South China Sea showed that the expected exhumed mantle is not present. If by hyper-extension you mean extreme extension of the continental crust, this is fine with me, but a more precise definition is needed.

2- Abstract, lines 29-32: a verb is missing in this sentence.

3- Line 41: "vigourous" ductile flow. Vigourous is probably too much here, intense shearing would better suit the observations.

4- Line 42: decoupling of upper and lower crust. This is only partly true in wide rift regions. In the Aegean, the Northern Tyrrhenian Sea or the Uruguay passive margin or the Afar, the direction of extension in the upper and lower crust are coaxial and it is true also for mantle fabrics suggesting that, while the detachments are partial decoupling zones, the whole lithosphere is sheared along the same direction with a likely control from mantle flow underneath (see my paper in Earth Sciences Reviews in 2018).

5- Lines 45-46: MCC do not always involve sediments. Several examples in the Aegean juxtapose low-pressure metamorphic rocks on top of higher-pressure (and higher temperature as well) without sediments.

6- Line 48: hyper-extension. Here the necessity of a clear definition of hyper-extension appears

again. The main characteristics of these “cold” continental margins is the exhumation of sub-continental mantle, not observed here.

7- Line 52: “flow of the lower crust balances upper crustal brittle faulting”. This is not always the case. There are examples in the Mediterranean and also the Uruguay margin where the lower crust is more extended than the upper crust and is thus exhumed in the OCT (see Gulf of Lion for instance).

8- Line 56: “hyperextended crust”. This is where the ambiguity is the most obvious. It is not the crust that is hyper-extended but the lithosphere, hence mantle exhumation. Hyper-extended crust should not be used here, in my view. It would blur the debate even more.

9- Some important papers are missing in the cited references, for instance Savva et al. (2013), one of the first clear evidence of crustal boudinage and ductile deformation of the South China Sea margin.

10- Line 128: I am fully convinced by the pure shear here. The images show a large displacement of the basin above the detachment and the folds observed in the southern deep part of the basin suggest to me the passage of these sediments on top of a flat-ramp-flat geometry of the detachment.

11- Lines 166-184. This paragraph is interesting and the interpretation correct, I think. This is the first time such details are imaged within a MCC. But the interpretation is not entirely clear. You do not say whether the low-reflectivity domains in the core of folds, which are clearly sheared by the activity of the detachment above, are plutonic intrusions or not. Their geometry is really similar to those we have observed in the Aegean in Serifos, Naxos or Ikaria.

12- Lines 203-205: a verb is missing here.

13- Lines 211-212: I do not understand here. Do you mean that intrusions emplaced only during the period characterized with pure shear ? This would not be compatible with the large deformation shown by the plutons.

14- Figure 1: please correct some typos: “apparent polarity” in the bottom of fig. 1b. “Volcanic construct” in the right side of the same figure.

15- Superb ! Figure 3b: I would interpret slightly differently the deep reflections on the right side. My impression is that we see south-dipping shear zones deflecting a slightly north-dipping foliation, one more argument in favor of top-south shearing.

16- Reference #30: this reference is totally wrong, the title is not complete and it is not a Geol. Soc. London special publication but a GSA special publication.

Laurent Jolivet

Reviewer #2 (Remarks to the Author):

The manuscript presents a spectacular new seismic dataset from the northern margin of the South China Sea and draws analogies between the structures observed there and in metamorphic core complexes. It is beautifully illustrated and generally well written, and will be of interest to a broad audience, so is appropriate for the journal. As far as I am aware, these structures have not been imaged before in a rifted margin setting. In some ways I am disappointed that there is not more quantitative analysis of the dataset, which has great potential, but the authors have developed a nice story that focuses on a specific aspect.

I have just two substantive comments:

1. There should be some comparison with similar imaging of the S-detachment on the Galicia margin (Schuba et al. <https://doi.org/10.1016/j.epsl.2018.04.012> and Lymer et al. <https://doi.org/10.1016/j.epsl.2019.03.018>), which also exhibits corrugations. The tectonic setting is a little different because S is more distal, but some comparison of dimensions and wavelengths of corrugations would be interesting.

2. In Figure 1b, the legend should be moved elsewhere – at present it obscures an important part of the seismic section. Since this part of the section is also obscured in Fig. S2, it may be that the authors do not have permission to show it. If that is the case, the corresponding part of the interpretation must also be removed, since no evidence is presented to support it. Such omission would not compromise the story developed in the paper.

Otherwise I have just a few detailed comments, keyed by line number:

30: Delete "that" – otherwise not a grammatical sentence.

50: Strictly the entire crust becomes brittle at 4-14 km thickness in ref. 14 – using one number is a bit too precise.

60-61: Not clear precisely what is meant here. There is well-documented exhumation of lower crustal rocks at, for example, the west Iberia margin – the MCC terminology is not normally used in this environment, but is that just because the crust is much thinner?

79: "later" -> "latter"

87-90: What is the evidence that the picked reflectors mark the boundary between the upper crust and the lower crust?

90-92: The sentence is missing a verb.

203-205: The sentence is missing a verb.

Fig. 2a: Mark also the location of the section in Fig. 1.

Fig. 3: State in the caption what the white vertical dashed lines represent.

Fig. 4c: One of the arrows from the word "corrugations" does not appear to point at corrugations.

Methods: The description of the data acquisition and processing is completely inadequate. I am sceptical that there were "two air guns" – perhaps two airgun arrays? Give the volume and tow depth of these arrays. State the number and spacing of streamers and the streamer length, depth and number of channels. Summarise the processing sequence and in particular how velocities were determined for depth migration.

Fig. S1: The caption should give some indication of the origin of this structural map – a citation if it is based on one that is published elsewhere, or some detail of how it was made if it is new.

Tim Minshull

Reviewer #3 (Remarks to the Author):

The manuscript: "Transition from wide continental rift to hyper-extended continental margin and to break-up north of the South China Sea" presents a very interesting and timely piece of work. The paper describes new 3D data in the northern sector of the South China sea, where the drilling took place. The authors interpret that the style of deformation is dominated by core-complexes and magmatic intrusion in a very hot and ductile environment and compares it to the structural styles observed in other wide extensional areas of the world which are characterized by core-complexes. The data is spectacular and the comparison to other onshore examples of core-complexes is good, pertinent and timely and fits well with the recent results of IODP drilling. In

terms of presentation, the Discussion and Conclusions are well written and the figures very good and illustrative.

I have, however, some questions about the authors interpretation which I hope will help improve the paper:

- I keep wondering if the Liwan detachment really worked as such, and it is not an angular unconformity formed during large-scale slumping of the sedimentary sequence above the detachment, generated as the margin was subsiding. In this sense, it would be good to see what is below the southern end of the LWDF (now it is covered by a legend in Figure 1b). If such a large continuous detachment, 100 km in the seismic section shown in Fig. 1a, would have been operative, it would have completely dissected the crust at its southern end, unless there was massive lateral lower crustal flow to fill in the extension produced by the detachment, perhaps combined with magmatic intrusions filling in the generated space and allowing the detachment not to break the crust? This is a situation which is different to the Aegean and North American cordillera because there the crust is quite thick at the time the detachment is active, so even if the detachment is large, lower crustal flow can compensate for that. Can the authors comment on what is the role of the detachment in crustal thinning? A figure which conceptually illustrates how the thinning produced by a 100 km long detachment would be compensated by lateral lower crustal flow in order to prevent break-up would be helpful. This could be incorporated in Figure 4. Actually, the color scale in that figure needs to be improved and the figure may be improved in general to show not only the similarities with other core-complex areas, but also how the core-complexes work in regimes where the crust is much thinner (note that the detachment in the South China sea seems to be active on a crust that is 10-15 km thick, and not 40 km or more like in the examples in the Aegean and North American cordillera).
- The Woodlark basin is also a highly extended basin where core-complexes have been observed. It would have been nice to compare those structures to the data shown here.
- What is the evidence that the age of the sedimentary sequences on top of the Liwan detachment are younging oceanward (Lines 121-131)? If the interpretation is based on existing wells, please cite.
- Is there is wide-angle data that can support the interpretation that there are magmatic intrusions in the area (I think there is).
- The first introductory paragraph (lines 63 to 64), needs more work to make the aims of the paper clearer, better differentiate margin extensional styles, and better recognize previous works which have pointed in the same direction as this one (see also detailed comments on particular sentences below). For example, it is clear that in the last 10 years the idea that all non-magma-rich margins would evolve in a similar way to magma-poor margins of the West Iberia-Newfoundland type (also ancient Tethys margins), which here I call magma-poor margins of the North-Atlantic type, has been strongly pushed by some authors. They have also coined the new term of "hyper-extension". While this is a 'sexy' term, it is almost void of significance, as all margins are hyper-extended (they all go to break-up and thus lithosphere thins to zero), and hence one cannot explain all "hyper-extended" margins using the same evolutionary model. Here this term is also used, contributing to blur the difference between "magma-poor margins of the North-Atlantic type" and the type of deformation described here for the South China sea, the North American Cordillera and the Aegean domain, while at the same time putting it in contrast. By using the term hyper-extension without further clarification it confuses the contribution of this work. I am not against the term hyper-extension, but without further clarification it is an unhelpful term. For example, one should describe what is meant by hyperextension, i.e. crust thinner than 15 km, and make the differentiation between wide hyper-extended crusts (i.e. South Atlantic, South China sea) and much narrower domains of hyper-extended crust (i.e. West Iberia, see for example, Ros et al., 2017, Brune et al., 2017, Huisman and Beaumont, 2011). It is clear that such contrasting margin widths cannot be generated by the same extensional processes nor the same initial conditions (see also earlier papers detailing the contrasting evolutions that margins would undergo in terms of melting and serpentinisation and type of break-up, Perez-Gussinye et al., 2001, Ros et al., 2017). Also, here it is ignored that the conceptual model for magma-poor margin evolution (of

the North-Atlantic type) which involves slow extension under cool conditions, coupling and embrittlement of the crust leading to serpentinisation, detachment formation and mantle exhumation was developed by Perez-Gussinye and Reston, 2001, and instead attributed to works that have incorporated these concepts on to an evolutionary framework that has been deemed to work for all margins, an idea that this work and many others disprove. Please cite properly. In terms of work in the South China sea itself, many papers including those coming from IODP results have already suggested that the deformation in South China sea would be different to that in the West-Iberia margins, and that the continent-ocean transition is abrupt and juxtaposes continental crust and magmatic oceanic crust, unlike in West Iberia (see Cameselle et al. 2017 for seismic evidence of this, and Sun et al., 2019, Larsen et al., 2018 showing IODP results). Please cite.

Detailed comments.

• Lines 46-49:

"MCCs forms in areas of high geothermal gradient^{5,8}, so it remains unclear how they relate to the more recent concept of hyperextended crust, that has been largely defined based on characteristic of cold continental margins^{11–13}. "

The tectonics of magma-poor cold continental margins have been described by many authors. Among them, Perez-Gussinye and Reston, 2001, firstly proposed that in these cold environments, dominated by very slow extension velocities, embrittlement of the crust would lead to the formation of crustal scale faults allowing mantle serpentinisation, detachment formation and crustal break-up, followed by mantle exhumation. These are margins dominated by large brittle structures and little lower crustal flow. They also calculated the stretching factors at which embrittlement and coupling would happen and compared it to observations. Perez-Gussinye et al., 2001, already showed that this type of evolution was restricted only to some margins, and that others, (such as the Woodlark basin, where core-complexes have been described), and which present higher initial geothermal gradients and rheological conditions, would exhibit more predominance of ductile extension and magmatism during extension and no mantle exhumation (see also Ros et al., 2017). Nevertheless, many authors later on, have taken these concepts i.e. coupling of the crust, embrittlement leading to serpentinisation etc, combined them with observations and attempted to apply them to all margins worldwide, as if one mode of tectonic evolution would fit all margins. Hence, I very much agree with the authors that the evolution of the South China sea is different from that of cold, magma-poor margins such as the West Iberia-Newfoundland, however I would encourage the authors to properly cite the primary source of many of the concepts developed for those margins (See also Perez-Gussinye et al., 2003, Perez-Gussinye et al., 2006, Ranero and Perez-Gussinye, 2010, Perez-Gussinye, 2013, and Ros et al., 2017, Brune et al., 2017).

• Lines 49-52: "Wide zones of hyperextended continental crust and their evolution to ocean-continent transition zones requires crustal thinning to ≤ 8 km leading to the exhumation of lower crust and mantle underneath low-angle detachments¹⁴."

I understand what the authors mean here, but the term hyperextended crust is very confusing and not well defined.... Hyperextended crust only means that the crust is very thin.... and the crust is always very thin before break-up. So I think the sentence above is confusing and paper 14 should be cited not as giving a number on crustal thickness and detachment formation, instead, on developing all these concepts, as they were unknown before. Please correct.

Papers 11-13 have branded the term "hyper-extended" to describe the distal domains of margins that do not exhibit Seaward Dipping Reflectors. While all margins have hyper-extended crust, the key difference between them is their width, with wide margins recording much more influence of ductile deformation processes and magmatism than narrow ones (see Ros et al., 2017).

I would recommend to clarify what authors mean by hyper-extension and differentiate between

wide and narrow margins, and proper citing of works that have gone in the direction of what is demonstrated with the images shown in this paper.

- Line 52-53: " Flow of the lower crust balances upper crustal brittle faulting which migrates oceanwards and accommodates thick sediment deposits >10 km"

I would appreciate if the authors could give a bit more detail of the margins they are talking about. At the moment the description for different margins is combined into a general description and it does not fit any real margin. For example, at the West-Iberia- Newfoundland margin the lower crust is not really being able to balance extension in the upper crust in the hyperextended domain, because it is too cold, it can not flow much.

Also, in the West Iberia margin the high velocity bodies have never been interpreted as magmatic intrusive bodies, but as serpentinised mantle. It has been in the Norwegian margin where the conceptual evolution of West Iberia-Newfoundland margins has been applied to interpret the high velocity lower crustal bodies as serpentinised mantle (although many authors still think they are magmatic intrusions).

- Line 58: " While research on wide continental rifting has focused on collapsed orogenic crust^{5,6} and studies on hyperextended crusts have focused on wide continent to ocean transitions^{12,13}, few studies have considered the possible link between the two.

Please cite studies that have actually related wide deformation mode with type of COT, melting and deformation structures (see Ros et al., G-cubed 2017, for a paper that makes this link).

- Line 202: hyperextended marginal crust

Following my comments above, I think the term "hyper-extension" is not well used here as it puts together under one umbrella margins which behave completely different, as the authors already recognize. The West Iberia-Newfoundland margins and the Alpine Tethys cannot be put under the same label as the South China sea margins. It is as if you would use a single word to describe all erosive and non-erosive subduction systems... If we keep doing this the community we are just producing endless loops of mis-information.

- Line 223: "stark contrast to the large-scale of mantle exhumations and usually one necking domain of cold continental margins"

Please cite properly the papers that are the primary source for accepted conceptual models in this type of margins. See above

Typos and minor comments

Few sentences are not properly built and the figure captions need to be double check. The authors talk about a figure S5 which does not exist, and the Supplementary figure numbers are sometimes S2, S3 and others 2S and 3S...

Some small comments are:

- Put distance ticks marks on x-axis in Fig. 1.
- Caption Figure 3. 'extension in the on the south of the detachment...'
- Caption Figure 3: " between T70–T70 "
- Line 16: correct typo: "continental lithosphere thins and break up.."
- Line 30: This sentence does not appear to be finished....

"The thermal and mechanical weakening of this broad continental domain that allowed for the

formation of metamorphic core complexes, boudinage of the upper crust and exhumation of middle/lower crust underneath a strongly boudinaged upper crust "

- Please check naming of lines A-A', B-B' in figure 2 and 3.
- Line 163: "the concave-upwards geometry of the detachment fault was likely locked up as extension and exhumation persist and was later replaced by the formation of new detachment fault in the hangingwall (Fig. 4a)39,40. " I think this refers to figure 3b, where the new detachment fault is shown.

References

Brune, Heine, Clift, Pérez-Gussinyé, Rifted margin architecture and crustal rheology: Reviewing Iberia-Newfoundland, Central South Atlantic, and South China Sea, *Marine and Pet. Geology.*, 10.1016/j.marpetgeo.2016.10.018, 2016

Cameselle, A. L., Ranero, C. R., Franke, D. and Barckhausen, U., 2-17, The continent-ocean transition on the northwestern South China Sea, *Basin Research* (2017) 29 (Suppl. 1), 73–95, doi: 10.1111/bre.12137

Huisman, R. S. & Beaumont, C. 2011. Depth-dependent extension, two-stage breakup and cratonic underplating at rifted margins. *Nature*, 473, 74 – 79, <http://dx.doi.org/10.1038/nature09988>

Pérez-Gussinyé, M., Ranero, C. R., Reston, T. J. & Sawyer, D. 2003. Mechanisms of extension at non-volcanic margin: evidence from the Galicia Interior Basin, west of Iberia. *Journal of Geophysical Research*, 108, 2245, <http://dx.doi.org/10.1029/2001JB000901>.

Pérez-Gussinyé, M., Phipps-Morgan, J., Reston, T. J. & Ranero, C. R. 2006. From rifting to spreading at non-volcanic margins: insights from numerical modeling. *Earth and Planetary Science Letters*, 244, 458–473.

Pérez-Gussinyé, M., 2013, A tectonic model for hyperextension at magma-poor rifted margins: an example from the West Iberia–Newfoundland conjugate margins, 2013, Geological Society, London, Special Publications 369 (1), 403-427.

Ranero & Pérez-Gussinyé, Sequential faulting explains the asymmetry and extension discrepancy of conjugate margins, *Nature*, 468, 294–299 2010.

Ros, E., Pérez-Gussinyé, M., Aarújo, M., Thoaldo Romeiro, M., Andres-Martinez, M., Morgan, JP, (2017), Lower crustal strength controls on melting and serpentinisation at magma-poor margins: potential implications for the South Atlantic, *G-cubed*, 18, 4538-4557, doi, 10.1002/2017GC007212.

Liheng Sun, Zhen Sun, Xiaolong Huang, Yingde Jiang and Joann Miriam Stock, Microstructures documenting Cenozoic extension processes in the northern continental margin of the South China Sea, *INTERNATIONAL GEOLOGY REVIEW*, <https://doi.org/10.1080/00206814.2019.1669079>

Point-by-point response to reviewers' comments

Hongdan Deng^{a,*}, Jianye Ren^a, Xiong Pang^b, Patrice Rey^c, Ken McClay^d, Ian Watkinson^e, Jingyun Zheng^b, Pan Luo^a

^a *College of Marine Science and Technology, China University of Geosciences, Wuhan, 430074, China*

^b *CNOOC Ltd. Shenzhen branch, Shenzhen, 518054, China*

^c *Earthbyte Research Group, Basin Genesis Hub, School of Geosciences, University of Sydney, Sydney, NSW 2006, Australia*

^d *Australian School of Petroleum, Adelaide University, North Terrace, Adelaide, SA 5000, Australia*

^e *SE Asia Research Group, Department of Earth Sciences, Royal Holloway University of London, Egham, UK*

* *Corresponding author (e-mail: denghongdan@gmail.com)*

Ref: NCOMMS-20-07610A

The comments prompted by the reviewers have been considered very carefully and are responded individually. In the following response, we first reply to Reviewer #1, and then to Reviewer #2 and Reviewer #3. The reviewers' comments are in italic.

In the resubmitted manuscript, all the changes are either marked in red (deleted) or highlighted with yellow background (added)

Response to Reviewer #1:

Thanks very much for the positive feedback from Professor Laurent Jolivet. We agree that this research will be very important for the understanding of wide

continent rifts development in the South China Sea and other rifted margins. It becomes clearer from our study that hot magma-poor continental margins are dominated by intense lower crustal extension coupled with upper crustal boudinage, and detachment faulting. This type of deformation explains the transition from wide continental rifting to break-up.

Specific comments 1:

Abstract: how do you define hyper-extension? This is ambiguous in the literature and it is often associated with mantle exhumation in the OCT. Clearly this is not the case here and recent drilling in the South China Sea showed that the expected exhumed mantle is not present. If by hyper-extension you mean extreme extension of the continental crust, this is fine with me, but a more precise definition is needed.

Reply:

We accept the reviewer's suggestion. The concept of hyper-extension, although was not explicitly defined in literature, was initially mentioned in typical cold magma-poor margins, such as the West Iberia-Newfoundland margins^{1,2}. These continental margins are characterised by crustal thickness of less than 14 km and by the presence of exhumed mantle^{3,4}. In contrast to the cold continental margins, the South China Sea is a typical hot margin without evidence of mantle exhumation^{5,6}. Therefore, the concepts of "hyper-extension" and "hyper-extended margin" can hardly be applicable in the South China Sea, and it could be very confusing to keep the same term in the description. Therefore, we now use "highly extended" to describe the continental crust that is less than 15 km in thickness in the South China Sea margins. Similar questions raised by other reviewers will be directed to this reply.

Specific comments 2:

Abstract, lines 29-32: a verb is missing in this sentence.

Reply:

We have removed “that” in Line 33.

Specific comments 3:

Line 41: “vigourous” ductile flow. Vigourous is probably too much here, intense shearing would better suit the observations.

Reply:

We have replaced “vigourous” by “intense” in Line 44.

Specific comments 4:

Line 42: decoupling of upper and lower crust. This is only partly true in wide rift regions. In the Aegean, the Northern Tyrrhenian Sea or the Uruguay passive margin or the Afar, the direction of extension in the upper and lower crust are coaxial and it is true also for mantle fabrics suggesting that, while the detachments are partial decoupling zones, the whole lithosphere is sheared along the same direction with a likely control from mantle flow underneath (see my paper in Earth Sciences Reviews in 2018).

Reply:

We agree with the reviewer’s comment. In some cases, such as examples provided by Jolivet et al. 2018⁷, deformations in the upper and lower crustal and even the mantle are not decoupled. We have, therefore, changed the sentence to “This mode

involves intense ductile flow of the lower crust and upper crustal boudinage, facilitated in many cases by the formation of detachment faults”. The changes are made in Lines 43–46.

Specific comments 5:

Lines 45-46: MCC do not always involve sediments. Several examples in the Aegean juxtapose low-pressure metamorphic rocks on top of higher-pressure (and higher temperature as well) without sediments.

Reply:

We agree with the reviewer’s comments. The presence and preservation of supradetachment sediments depend on sediments supply and depositional environment. In cases of sediment starvation and strong erosion, we may not see sediments in the MCC system. We have replaced “young sediments over exhumed” by “upper crustal, brittlely deformed rocks with lower crustal, ductilely deformed” in Lines 48–49.

Specific comments 6:

Line 48: hyper-extension. Here the necessity of a clear definition of hyper-extension appears again. The main characteristics of these “cold” continental margins is the exhumation of sub-continental mantle, not observed here.

Reply:

We agree with the reviewer’s comments. Similar reply to this comment can be found in the reply to “**specific comment 1**”. The term of “hyper-extension” has been removed throughout the manuscript.

Specific comments 7:

Line 52: “flow of the lower crust balances upper crustal brittle faulting”. This is not always the case. There are examples in the Mediterranean and also the Uruguay margin where the lower crust is more extended than the upper crust and is thus exhumed in the OCT (see Gulf of Lion for instance).

Reply:

We agree with the reviewer’s comment and have removed the sentence “Flow of the lower crust balances upper crustal brittle faulting which migrates oceanwards and accommodates thick sediment deposits >10 km”.

Specific comments 8:

Line 56: “hyperextended crust”. This is where the ambiguity is the most obvious. It is not the crust that is hyper-extended but the lithosphere, hence mantle exhumation. Hyper-extended crust should not be used here, in my view. It would blur the debate even more.

Reply:

We agree with the reviewer’s comments. Similar reply to this comment can be found in the reply of “**specific comment 1**”.

Specific comments 9:

Some important papers are missing in the cited references, for instance Savva et al. (2013), one of the first clear evidence of crustal boudinage and ductile deformation of the South China Sea margin.

Reply:

We have added Savva et al. (2013) – Seismic evidence of hyper-stretched crust and mantle exhumation offshore Vietnam – as a reference.

Specific comments 10:

Line 128: I am fully convinced by the pure shear here. The images show a large displacement of the basin above the detachment and the folds observed in the southern deep part of the basin suggest to me the passage of these sediments on top of a flat-ramp-flat geometry of the detachment.

Reply:

Thanks for the supportive comment.

Specific comments 11:

Lines 166-184. This paragraph is interesting and the interpretation correct, I think. This is the first time such details are imaged within a MCC. But the interpretation is not entirely clear. You do not say whether the low-reflectivity domains in the core of folds, which are clearly sheared by the activity of the detachment above, are plutonic intrusions or not. Their geometry is really similar to those we have observed in the Aegean in Serifos, Naxos or Ikaria.

Reply:

Thanks for the reviewer's comment. We now state clear that the low-reflectivity domains in the core of folds are tilted plutonic intrusions similar to granitoid-cored MCC in Serifos, Naxos, and Ikaria^{8,9}. The change is made in Lines 202-203.

Specific comments 12:

Lines 203-205: a verb is missing here.

Reply:

We have added a “.” in Line 225.

Specific comments 13:

Lines 211-212: I do not understand here. Do you mean that intrusions emplaced only during the period characterized with pure shear? This would not be compatible with the large deformation shown by the plutons.

Reply:

No, we mean that the later-stage pure-shear extension occurred together with intrusion emplacement. These intrusions are formed the D1, D2, and D3. Obviously, the dominant deformation is simple-shear deformation that transposed the top of the pluton southward for more than 20 km. To make this sentence clearer, we changed the sentence to “the LWDF was chiefly controlled by simple-shear deformation in Eocene extension and by localised pure-shear deformation in Oligocene extension, the later coeval with deep crust intrusion and strain localization during continental break-up”. The change can be found in Lines 232-235.

Specific comments 14:

Figure 1: please correct some typos: “apparent polarity” in the bottom of fig. 1b. “Volcanic construct” in the right side of the same figure.

Reply:

Thanks for pointing out this. We have corrected the typos “apparent polarity” in Figure 1 and Figure 3 and “volcanic construct” in Figure 1b.

Specific comments 15:

Superb ! Figure 3b: I would interpret slightly differently the deep reflections on the right side. My impression is that we see south-dipping shear zones deflecting a slightly north-dipping foliation, one more argument in favor of top-south shearing.

Reply:

Thanks for the supportive comment. For the interpretation, I think it does not make a big difference if we interpret the deep reflection slightly on the right side. Our interpretation is consistent with the reflection data and seismic attribute data.

Specific comments 16:

Reference #30: this reference is totally wrong, the title is not complete and it is not a Geol. Soc. London special publication but a GSA special publication.

Reply:

Thanks for pointing out this. We have corrected this reference information. The correct citation can be found in reference #41.

Response to Reviewer #2:

Thanks for the reviewer's positive feedback on our manuscript. We have added quantitative analysis of the dataset. This includes the orientation plot of secondary faults in Table-1 and orientation plot of grooves in Table-2. Apart from that we also added the aspect ratio and dimensions of 49 corrugations or MCC domes of the North America Cordilleran, Aegean, Papua New Guinea, North China, southern France, Philippine Sea, and West Iberian margin (Table-3). These aspect ratios of metamorphic core complexes, oceanic core complexes, and detachment corrugations are compared with corrugations and sub-domes of our study in the South China Sea. The plots are shown in Figure 4.

Specific comments 1:

There should be some comparison with similar imaging of the S-detachment on the Galicia margin (Schuba et al. <https://doi.org/10.1016/j.epsl.2018.04.012> and Lymer et al. <https://doi.org/10.1016/j.epsl.2019.03.018>), which also exhibits corrugations. The tectonic setting is a little different because S is more distal, but some comparison of dimensions and wavelengths of corrugations would be interesting.

Reply:

We have accepted the reviewer's suggestion and have compared corrugations of S-detachment on the Galicia margin with corrugations^{10,11} and domes of the Liwan detachment in our study. Essentially, the S-detachment or S-reflector is a serpentinite detachment with serpentinitized mantle in the footwall⁴, whereas the Liwan detachment has middle and lower crust in its footwall. The dimension and aspect ratio of these corrugations and metamorphic core complexes and oceanic

core complexes were shown in Figure 4. It is obvious that the corrugations of S reflector on the Galicia margin (Fig. 4b) has smaller dimensions (<8 km in length and <1.5 km in width) than others. The aspect ratio of the corrugations of S reflector fills the range of 6 to 16, much larger than the aspect ratio of most metamorphic complexes. The differences in size and in aspect ratio of hot (South China Sea) and cold (Galicia) continental margins may suggest that temperature plays a significant role in shaping the geometry of MCC corrugation and dome of detachment faults. We have added more in Lines 160-167.

Specific comments 2:

In Figure 1b, the legend should be moved elsewhere – at present it obscures an important part of the seismic section. Since this part of the section is also obscured in Fig. S2, it may be that the authors do not have permission to show it. If that is the case, the corresponding part of the interpretation must also be removed, since no evidence is presented to support it. Such omission would not compromise the story developed in the paper.

Reply:

The lower part of the seismic section is not provided from the seismic data we are using. We have removed the interpretation of this area in Figure 1c.

Specific comments 3:

Line 30: Delete “that” – otherwise not a grammatical sentence.

Reply:

We accept the reviewer's suggestion and have removed "that" in Line 33. Also see our reply to "***Specific comments 2***" of #Reviewer 1.

Specific comments 4:

Line 50: Strictly the entire crust becomes brittle at 4-14 km thickness in ref. 14 – using one number is a bit too precise.

Reply:

We have removed this sentence.

Specific comments 5:

Lines 60-61: Not clear precisely what is meant here. There is well-documented exhumation of lower crustal rocks at, for example, the west Iberia margin – the MCC terminology is not normally used in this environment, but is that just because the crust is much thinner?

Reply:

The west Iberian margin, which has been extensively studied in the last twenty years, is a typical cold margin^{1,4,12-14}. It is characterised by mantle exhumation and serpentinization. We think that the corrugations of S reflector, as mentioned in ***specific comments 1***, is different from typical metamorphic core complexes in hot continental rifts or rifted margins such as the North America Cordilleran, Argean, Paupa New Guinea etc (see the statistic work of MCCs in Table-3).

Specific comments 6:

Line 79: “later” -> “latter”

Reply:

We have changed “later” to “latter” in Line 90.

Specific comments 7:

Line 87-90: What is the evidence that the picked reflectors mark the boundary between the upper crust and the lower crust?

Reply:

This boundary is a continuous reflector in the crust (yellow dots in Figure 1b). Most brittle faults stopped at this reflector. Therefore, we infer that this reflector represents the brittle/ductile transition within the crust. The crust thickness over this reflector, based on the velocity 6 km/s (crystalline basement), is about 12 km, and therefore the thickness fit well into the brittle/ductile transition zone commonly observed at 10-15 km interval¹⁵.

Specific comments 8:

Lines 90-92: The sentence is missing a verb.

Reply:

We have added “is” in Line 101.

Specific comments 9:

203-205: The sentence is missing a verb.

Reply:

We have replaced “.” by “:” in Line 225.

Specific comments 10:

Fig. 2a: Mark also the location of the section in Fig. 1.

Reply:

We have marked the section location in Figure 2.

Specific comments 11:

Fig. 3: State in the caption what the white vertical dashed lines represent.

Reply:

We have stated in the caption that the white vertical dashed lines represent dome axis in cross-section.

Specific comments 12:

Fig. 4c: One of the arrows from the word “corrugations” does not appear to point at corrugations.

Reply:

All the arrows point to the word “corrugations” in Figure 5c. The left arrow points to a faulted corrugation anticline, whereas the right arrow points to a non-faulted corrugation anticline.

Specific comments 13:

Methods: The description of the data acquisition and processing is completely inadequate. I am sceptical that there were “two air guns” – perhaps two airgun arrays? Give the volume and tow depth of these arrays. State the number and spacing of streamers and the streamer length, depth and number of channels. Summarise the processing sequence and in particular how velocities were determined for depth migration.

Reply:

Thanks for pointing this out. We have now provided more information on seismic data acquisition and processing, as following –

The 3D seismic dataset was acquired by CNOOC in 2011 using two airgun arrays spaced 50 m apart, fired every 25 m at 2000 psi. The total volume used was 7750 in³, towing at depth of 6±0.5 m throughout the survey. The data were collected using twelve hydrophone streamers, each 600 m in length, containing 480 channels, regularly spaced at 12.5 m. The data were recorded for 8192 ms at a 1-ms sample rate and were reprocessed and generated pre-stack depth migration data by PGS in 2012. It has an inline (N–S) and crossline (E–W) spacing of 12.5 m * 12.5 m, 5 m vertical sample rate, covers an area of ~1500 km² and has a record length of 10 km. There are 3142 inlines and 3816 crosslines in the data volume. The main frequency of the reprocessed volume is 30-45 Hz. The velocity model applied for the depth conversion of the seismic volume is consistent with the velocity from ocean bottom seismic OBS93¹⁶.

Specific comments 14:

Fig. S1: The caption should give some indication of the origin of this structural map – a citation if it is based on one that is published elsewhere, or some detail of how it was made if it is new.

Reply:

Thanks for pointing this out. The structural map is modified from Xie et al., (2019)¹⁷. We have added the citation (#31 reference) in the figure caption of Fig. S1.

Response to Reviewer #3:

Thanks very much for the positive feedback on our manuscript. We appreciate the effort of the reviewer for his/her insightful comments and suggestions that expand our contribution to wider audiences who concerned about 'hot' and 'cold' continental margins evolution worldwide.

Specific comments 1:

I keep wondering if the Liwan detachment really worked as such, and it is not an angular unconformity formed during large-scale slumping of the sedimentary sequence above the detachment, generated as the margin was subsiding. In this sense, it would be good to see what is below the southern end of the LWDF (now it is covered by a legend in Figure 1b). If such a large continuous detachment, 100 km in the seismic section shown in Fig. 1a, would have been operative, it would have completely dissected the crust at its southern end, unless there was massive lateral lower crustal flow to fill in the extension produced by the detachment, perhaps combined with magmatic intrusions filling in the generated space and allowing the detachment not to break the crust? This is a situation which is different to the Aegean and North American cordillera because there the crust is quite thick at the time the detachment is active, so even if the detachment is large, lower crustal flow can compensate for that. Can the authors comment on what is the role of the detachment in crustal thinning? A figure which conceptually illustrates how the thinning produced by a 100 km long detachment would be compensated by lateral lower crustal flow in order to prevent break-up would be helpful. This could be incorporated in Figure 4. Actually, the color scale in that figure needs to be improved and the figure may be improved in general to show not only the similarities with other

core-complex areas, but also how the core-complexes work in regimes where the crust is much thinner (note that the detachment in the South China sea seems to be active on a crust that is 10-15 km thick, and not 40 km or more like in the examples in the Aegean and North American cordillera).

Reply:

Unfortunately, the seismic section below the southern end of the Liwan detachment fault (Fig. 1b) is not available in our study. Similar answer has been replied to “**Specific comments 2**” of #Reviewer 2. This seismic section is located next to a deep seismic section extensively used in IODP⁵ that imaged the lower crust and upper mantle (Fig. 2a in Larsen et al., 2018 Nature Geoscience). In the paper of Larsen et al. (2018), the Liwan detachment does not dissect the crust but flattens out to the surface. It may have a direct contact with oceanic crust in further south, similar to the sharp COT of Woodlark basin¹⁸. Indeed, the lower crustal flow combined with magmatic intrusions underneath the Liwan detachment were massive. This is supported by the top-to-the-south tilting of plutons (low, homogeneous amplitude) and south-verging nappe folds (Fig. 3) that have been transposed to the south for at least 20 km.

Detachment faulting in our study significantly thinned the continental upper crust and exhumed the middle and lower crust. So, basically the Liwan detachment removed the brittle layer of the continental crust, while the middle and lower crust remain unbroken. As the detachment moves southward, weak and ductile lower crust and plutonic intrusions migrate into the footwall. The flow of ductile material towards the footwall of the Liwan detachment is similar to rolling-hinge and sequential faulting style of deformation^{12,19}. A conceptual model of the highly extended continental crust

of the northern South China Sea has been provided in Figure 5C to show that the 100 km upper crustal detachment faulting has been compensated by oceanward lower crustal flow.

Specific comments 2:

The Woodlark basin is also a highly extended basin where core-complexes have been observed. It would have been nice to compare those structures to the data shown here.

Reply:

It is true that the Woodlark basin is a highly extended basin with metamorphic core complexes observed near current tip of the Woodlark spreading ridge²⁰. Both the Woodlark basin and the South China Sea have hot continental margins with metamorphic core complexes development. We have taken the reviewer's comments and compared these two rift basins in terms of Rheology, thermal, and magmatic conditions¹⁸. However, contrast to the South China Sea, where the highly extended continental crust is <10 km, the Woodlark basin has more than 20 km of crust thickness at the place of metamorphic core complex development^{20,21}. So, it seems that the Woodlark basin has not experienced a stage of extremely thinning before break-up.

Specific comments 3:

What is the evidence that the age of the sedimentary sequences on top of the Liwan detachment are younging oceanward (Lines 121-131)? If the interpretation is based on existing wells, please cite.

Reply:

This is based on sedimentary stacking pattern of the Lower Eocene to Lower Oligocene sediments (Fig. 3). It is obvious that the Lower Eocene sediments are mainly accumulated in the north and the Upper Eocene and Oligocene sediments are progressively migrated to the south.

Specific comments 4:

Is there is wide-angle data that can support the interpretation that there are magmatic intrusions in the area (I think there is).

Reply:

Thanks for pointing this out. Indeed, there is a wide-angle ocean bottom seismic line, OBS-93²², across the study that shows volcanic composition in the footwall of the Liwan detachment. We have cited this information (#36 reference) to support our argument of magmatism in the footwall of the Liwan detachment in Lines 204-206.

Specific comments 5:

The first introductory paragraph (lines 63 to 64), needs more work to make the aims of the paper clearer, better differentiate margin extensional styles, and better recognize previous works which have pointed in the same direction as this one (see also detailed comments on particular sentences below). For example, it is clear that in the last 10 years the idea that all non-magma-rich margins would evolve in a similar way to magma-poor margins of the West Iberia-Newfoundland type (also ancient Tethys margins), which here I call magma-poor margins of the North-Atlantic type, has been strongly pushed by some authors. They have also coined the new

term of "hyper-extension". While this is a 'sexy' term, it is almost void of significance, as all margins are hyper-extended (they all go to break-up and thus lithosphere thins to zero), and hence one cannot explain all "hyper-extended" margins using the same evolutionary model. Here this term is also used, contributing to blur the difference between "magma-poor margins of the North-Atlantic type" and the type of deformation described here for the South China sea, the North American Cordillera and the Aegean domain, while at the same time putting it in contrast. By using the term hyper-extension without further clarification it confuses the contribution of this work. I am not against the term hyper-extension, but without further clarification it is an unhelpful term. For example, one should describe what is meant by hyperextension, i.e. crust thinner than 15 km, and make the differentiation between wide hyper-extended crusts (i.e. South Atlantic, South China sea) and much narrower domains of hyper-extended crust (i.e. West Iberia, see for example, Ros et al., 2017, Brune et al., 2017, Huisman and Beaumont, 2011). It is clear that such contrasting margin widths cannot be generated by the same extensional processes nor the same initial conditions (see also earlier papers detailing the contrasting evolutions that margins would undergo in terms of melting and serpentinisation and type of break-up, Perez-Gussinye et al., 2001, Ros et al., 2017). Also, here it is ignored that the conceptual model for magma-poor margin evolution (of the North-Atlantic type) which involves slow extension under cool conditions, coupling and embrittlement of the crust leading to serpentinisation, detachment formation and mantle exhumation was developed by Perez-Gussinye and Reston, 2001, and instead attributed to works that have incorporated these concepts on to an evolutionary framework that has been deemed to work for all margins, an idea that this work and many others disprove. Please cite properly. In terms of work in the

South China sea itself, many papers including those coming from IODP results have already suggested that the deformation in South China sea would be different to that in the West-Iberia margins, and that the continent-ocean transition is abrupt and juxtaposes continental crust and magmatic oceanic crust, unlike in West Iberia (see Cameselle et al. 2017 for seismic evidence of this, and Sun et al., 2019, Larsen et al., 2018 showing IODP results). Please cite.

Reply:

This comment by the reviewer is very insightful. Indeed, there are two styles of non-volcanic (magma-poor) margins. One is exemplified by the Iberia-Newfoundland conjugates or North Atlantic margins; the other is the South Atlantic margins^{18,23}. The Iberia and Newfoundland margins are classic magma-poor or non-volcanic margins that are characterised by serpentinite detachment or S reflector between the highly extended crust (<15 km) and serpentinitized mantle at the continent-ocean transition zone^{1,4,14,18}. The break-up of this type of continent margins involves mantle exhumation and coupled deformation. However, the South China Sea, the Woodwalk basin, South Atlantic margin, the North America Cordillera and Aegean areas are also non-volcanic margins but show strikingly different crustal deformation. These margins have no evidence of mantle exhumation and embrittlement of the crust in the continent-ocean transition zone. The difference of North Atlantic margins and the South China Sea margins is that the former is cold whereas the latter is hot margins.

Specific comments 6:

Lines 46-49: MCCs forms in areas of high geothermal gradient^{5,8}, so it remains unclear how they relate to the more recent concept of hyperextended crust, that has been largely defined based on characteristic of cold continental margins¹¹⁻¹³.

The tectonics of magma-poor cold continental margins have been described by many authors. Among them, Perez-Gussinye and Reston, 2001, firstly proposed that in these cold environments, dominated by very slow extension velocities, embrittlement of the crust would lead to the formation of crustal scale faults allowing mantle serpentinisation, detachment formation and crustal break-up, followed by mantle exhumation. These are margins dominated by large brittle structures and little lower crustal flow. They also calculated the stretching factors at which embrittlement and coupling would happen and compared it to observations. Perez-Gussinye et al., 2001, already showed that this type of evolution was restricted only to some margins, and that others, (such as the Woodlark basin, where core-complexes have been described), and which present higher initial geothermal gradients and rheological conditions, would exhibit more predominance of ductile extension and magmatism during extension and no mantle exhumation (see also Ros et al., 2017). Nevertheless, many authors later on, have taken these concepts i.e. coupling of the crust, embrittlement leading to serpentinisation etc, combined them with observations and attempted to apply them to all margins worldwide, as if one mode of tectonic evolution would fit all margins. Hence, I very much agree with the authors that the evolution of the South China sea is different from that of cold, magma-poor margins such as the West Iberia-Newfoundland, however I would encourage the authors to properly cite the primary source of many of the concepts developed for those margins (See also Perez-Gussinye et al., 2003, Perez-Gussinye et al., 2006, Ranero and Perez-Gussinye, 2010, Perez-Gussinye, 2013, and Ros et al., 2017, Brune et al., 2017).

Reply:

Thanks for pointing this out. Sorry that we missed some key papers on “cold” and “hot” margins. We have now cited all these papers in our manuscript.

Specific comments 7:

Lines 49-52: "Wide zones of hyperextended continental crust and their evolution to ocean-continent transition zones requires crustal thinning to ≤ 8 km leading to the exhumation of lower crust and mantle underneath low-angle detachments¹⁴."

I understand what the authors mean here, but the term hyperextended crust is very confusing and not well defined.... Hyperextended crust only means that the crust is very thin.... and the crust is always very thin before break-up. So I think the sentence above is confusing and paper 14 should be cited not as giving a number on crustal thickness and detachment formation, instead, on developing all these concepts, as they were unknown before. Please correct. Papers 11-13 have branded the term "hyper-extended" to describe the distal domains of margins that do not exhibit Seaward Dipping Reflectors. While all margins have hyper-extended crust, the key difference between them is their width, with wide margins recording much more influence of ductile deformation processes and magmatism than narrow ones (see Ros et al., 2017). I would recommend to clarify what authors mean by hyper-extension and differentiate between wide and narrow margins, and proper citing of works that have gone in the direction of what is demonstrated with the images shown in this paper.

Reply:

Thanks for pointing this out. Similar to the comments by #Reviewer1, we have removed the term “hyper-extension” as this term is most used for the description of “cold” continental margins.

Specific comments 8:

Line 52-53: " Flow of the lower crust balances upper crustal brittle faulting which migrates oceanwards and accommodates thick sediment deposits >10 km"

I would appreciate if the authors could give a bit more detail of the margins they are talking about. At the moment the description for different margins is combined into a general description and it does not fit any real margin. For example, at the West-Iberia- Newfoundland margin the lower crust is not really being able to balance extension in the upper crust in the hyperextended domain, because it is too cold, it can not flow much. Also, in the West Iberia margin the high velocity bodies have never been interpreted as magmatic intrusive bodies, but as serpentinitised mantle. It has been in the Norwegian margin where the conceptual evolution of West Iberia- Newfoundland margins has been applied to interpret the high velocity lower crustal bodies as serpentinitised mantle (although many authors still think they are magmatic intrusions).

Reply:

Thanks for the reviewer's comments on other margins. We have removed this sentence.

Specific comments 9:

Line 58: " While research on wide continental rifting has focused on collapsed orogenic crust^{5,6} and studies on hyperextended crusts have focused on wide continent to ocean transitions^{12,13}, few studies have considered the possible link between the two.

Please cite studies that have actually related wide deformation mode with type of COT, melting and deformation structures (see Ros et al., G-cubed 2017, for a paper that makes this link).

Reply:

We will cite the paper by Ros et al., G-cubed 2017 (#reference 21)

Specific comments 10:

Line 202: hyperextended marginal crust

Following my comments above, I think the term "hyper-extension" is not well used here as it puts together under one umbrella margins which behave completely different, as the authors already recognize. The West Iberia-Newfoundland margins and the Alpine Tethys cannot be put under the same label as the South China sea margins. It is as if you would use a single word to describe all erosive and non-erosive subduction systems... If we keep doing this the community we are just producing endless loops of mis-information.

Reply:

Thanks for the comment. We have removed the term 'hyper-extension' in the paper. Similar answer has been provided earlier.

Specific comments 11:

Line 223: "stark contrast to the large-scale of mantle exhumations and usually one necking domain of cold continental margins"

Please cite properly the papers that are the primary source for accepted conceptual models in this type of margins. See above

Reply:

We will cite the papers on cold magma-poor continental margins of the West Iberia-Newfoundland.

Specific comments 12:

Typos and minor comments

Few sentences are not properly built and the figure captions need to be double check. The authors talk about a figure S5 which does not exist, and the Supplementary figure numbers are sometimes S2, S3 and others 2S and 3S...

Reply:

Sorry about the inconsistent usage of figure numbers... We have now labelled the supplementary figures as S1, S2, S3, S4. We do not have S5 in the supplementary figures and have removed the label of S5 in Figure S4.

Specific comments 13:

Put distance ticks marks on x-axis in Fig. 1.

Reply:

We have put distance ticks on the x-axis in Fig.1.

Specific comments 14:

Caption Figure 3. 'extension in the on the south of the detachment...'

Reply:

We have removed “on the” in the sentence of ‘extension in the on the south of the detachment...’ in caption Figure 3.

Specific comments 15:

Caption Figure 3: " between T70–T70 "

Reply:

We have the replaced “T70-T70” by “T70-T60” in caption Figure 3.

Specific comments 16:

Line 16: correct typo: "continental lithosphere thins and break up.."

Reply:

We have replaced “break up” by “break-up”.

Specific comments 17:

Line 30: This sentence does not appear to be finished....

"The thermal and mechanical weakening of this broad continental domain that allowed for the formation of metamorphic core complexes, boudinage of the upper crust and exhumation of middle/lower crust underneath a strongly boudinaged upper crust "

Reply:

We have removed "that" in this sentence in Line 30. The same problem pointed out by #Reviewer1 and #Reviewer2 have been corrected.

Specific comments 18:

Please check naming of lines A-A', B-B' in figure 2 and 3.

Reply:

We have added A-A' and B-B' on top of the corresponding seismic sections in figure 3. In figure 2, the line location of A-A' and B-B' has already been annotated in the perspective view (Fig. 2a).

Specific comments 19:

Line 163: "the concave-upwards geometry of the detachment fault was likely locked up as extension and exhumation persist and was later replaced by the formation of new detachment fault in the hangingwall (Fig. 4a)39,40. " I think this refers to figure 3b, where the new detachment fault is shown.

Reply:

No, this sentence compares the rolling-hinge model of the Colorado River core complex (Fig. 5a) with the break-away of the Liwan detachment (Fig. 1b-c). It is obvious that the back-rotated fault block and bowed upwards fault geometry in the footwall of the Liwan detachment are inactive structures that have displacement taken up by new faults in the hangingwall (Fig. 1b-c). We agree with the reviewer's comment that in the middle part of the Liwan detachment (Fig. 3b), the fault was also locked up because of lower crust extrusion. As a consequence, a new fault developed in the footwall of the original detachment. However, the mechanism of footwall detachment fault evolution is different to that of the Colorado River detachment fault.

References

1. Peron-Pinvidic, G., Manatschal, G. & Osmundsen, P. T. Structural comparison of archetypal Atlantic rifted margins: A review of observations and concepts. *Mar. Pet. Geol.* **43**, 21–47 (2013).
2. Pérez-Gussinyé, M. A tectonic model for hyperextension at magma-poor rifted margins: an example from the West Iberia–Newfoundland conjugate margins. *Geol. Soc. London, Spec. Publ.* **369**, 403–427 (2013).
3. Buck, W. R. The Dynamics of Continental Breakup and Extension. in *Treatise on Geophysics* 325–379 (Elsevier, 2015). doi:10.1016/B978-0-444-53802-4.00118-4
4. Pérez-Gussinyé, M. & Reston, T. J. Rheological evolution during extension at nonvolcanic rifted margins: Onset of serpentinization and development of detachments leading to continental breakup. *J. Geophys. Res. Solid Earth* **106**,

- 3961–3975 (2001).
5. Larsen, H. C. *et al.* Rapid transition from continental breakup to igneous oceanic crust in the South China Sea. *Nat. Geosci.* **11**, 782–789 (2018).
 6. Larsen, H. C. *et al.* Expedition 367/368 summary. in **367**, (2018).
 7. Jolivet, L. *et al.* Extensional crustal tectonics and crust-mantle coupling, a view from the geological record. *Earth-Science Rev.* **185**, 1187–1209 (2018).
 8. Rabillard, A. *et al.* Synextensional Granitoids and Detachment Systems Within Cycladic Metamorphic Core Complexes (Aegean Sea, Greece): Toward a Regional Tectonomagmatic Model. *Tectonics* **37**, 2328–2362 (2018).
 9. Rabillard, A. *et al.* Interactions between plutonism and detachments during metamorphic core complex formation, Serifos Island (Cyclades, Greece). *Tectonics* **34**, 1080–1106 (2015).
 10. Schuba, C. N. *et al.* A low-angle detachment fault revealed: Three-dimensional images of the S-reflector fault zone along the Galicia passive margin. *Earth Planet. Sci. Lett.* **492**, 232–238 (2018).
 11. Lymer, G. *et al.* 3D development of detachment faulting during continental breakup. *Earth Planet. Sci. Lett.* **515**, 90–99 (2019).
 12. Ranero, C. R. & Pérez-Gussinyé, M. Sequential faulting explains the asymmetry and extension discrepancy of conjugate margins. *Nature* **468**, 294–299 (2010).
 13. Brune, S., Heine, C., Clift, P. D. & Pérez-Gussinyé, M. Rifted margin architecture and crustal rheology: Reviewing Iberia-Newfoundland, Central South Atlantic, and South China Sea. *Mar. Pet. Geol.* **79**, 257–281 (2017).
 14. Peron-Pinvidic, G. & Manatschal, G. Rifted Margins: State of the Art and Future Challenges. *Front. Earth Sci.* **7**, 1–8 (2019).

15. Sibson, R. H. Fault rocks and fault mechanisms. *J. Geol. Soc. London.* **133**, 191–213 (1977).
16. Huang, C., Zhou, D., Sun, Z., Chen, C. & Hao, H. Deep crustal structure of Baiyun Sag, northern South China Sea revealed from deep seismic reflection profile. *Chinese Sci. Bull.* **50**, 1131 (2005).
17. Xie, X., Ren, J., Pang, X., Lei, C. & Chen, H. Stratigraphic architectures and associated unconformities of Pearl River Mouth basin during rifting and lithospheric breakup of the South China Sea. *Mar. Geophys. Res.* **40**, 129–144 (2019).
18. Pérez-Gussinyé, M., Reston, T. J. & Morgan, J. P. Serpentinization and magmatism during extension at non-volcanic margins: The effect of initial lithospheric structure. *Geol. Soc. Spec. Publ.* **187**, 551–576 (2001).
19. Roger, B. W. Flexural Rotation of Normal Faults. *Tectonics* **7**, 959–973 (1988).
20. Little, T. A. *et al.* Diapiric exhumation of Earth's youngest (UHP) eclogites in the gneiss domes of the D'Entrecasteaux Islands, Papua New Guinea. *Tectonophysics* **510**, 39–68 (2011).
21. Little, T. A., Baldwin, S. L., Fitzgerald, P. G. & Monteleone, B. Continental rifting and metamorphic core complex formation ahead of the Woodlark spreading ridge, D'Entrecasteaux Islands, Papua New Guinea. *Tectonics* **26**, n/a-n/a (2007).
22. Pin, Y., Di, Z. & Zhaoshu, L. A crustal structure profile across the northern continental margin of the South China sea. *Tectonophysics* **338**, 1–21 (2001).
23. Huisman, R. & Beaumont, C. Depth-dependent extension, two-stage breakup and cratonic underplating at rifted margins. *Nature* **473**, 74–78 (2011).

REVIEWERS' COMMENTS:

Reviewer #2 (Remarks to the Author):

The authors have done a very thorough job of addressing reviewer comments, and I suggest just a few minor corrections, keyed by line number below.

54: "is dominating" -> "dominates"

55: "continental-ocean" -> "continent-ocean" (twice)

165: "fills into" -> "lies within"

166: "is different" -> "are different"

168: "has smaller" -> "have smaller"

171: "dome" -> "doming"

210-211: Rephrase as "The presence of volcanic material has been inferred previously from wide-angle seismic data".

266-267: The statement here is not really true – the measurements are tabulated in the supplementary material but the seismic data used are not there (nor would I expect them to be, since they are proprietary data).

Figure 2 caption: The supplementary material where the orientation data are given should be cited here.

Fig. 4 caption: "Dimension and aspect ratio" -> "Dimensions and aspect ratios". "S reflector" -> "the S reflector" and later "reflectors" -> "reflector" (twice). The caption should also cite the supplementary material where the origin of the measurements is given.

Tim Minshull

Reviewer #3 (Remarks to the Author):

The authors have responded satisfactorily all questions made. The additional comparison with the S reflector is very good and the introduction now reads much better. I think the manuscript should be published as soon as possible. I have no further comments to add.

South China Sea documents the transition from wide continental rift to continental break up

Hongdan Deng^{1,2*}, Jianye Ren^{2,3}, Xiong Pang⁴, Patrice Rey⁵, Ken McClay⁶, Ian Watkinson⁷, Jingyun Zheng⁴, Pan Luo²

¹ *Hubei Key Laboratory of Marine Geological Resources, China University of Geosciences, 430074, China*

² *College of Marine Science and Technology, China University of Geosciences, Wuhan, 430074, China*

³ *Key laboratory of Tectonics and Petroleum Resources of Ministry of Education, China University of Geosciences, Wuhan, China*

⁴ *CNOOC Ltd. Shenzhen branch, Shenzhen, 518054, China*

⁵ *Earthbyte Research Group, Basin Genesis Hub, School of Geosciences, The University of Sydney, Sydney, NSW 2006, Australia*

⁶ *Australian School of Petroleum, Adelaide University, North Terrace, Adelaide, SA 5000, Australia*

⁷ *SE Asia Research Group, Department of Earth Sciences, Royal Holloway University of London, Egham, UK*

** Correspondence: denghongdan@gmail.com*

Ref: NCOMMS-20-07610B

The comments prompted by the 2nd reviewer have been considered very carefully and are responded individually. The reviewer' comments are in italic.

Specific comments 1:

54: “is dominating” -> “dominates”

Reply:

We accept the reviewer’s suggestion and have changed “is dominating” to “dominates” in Line 55.

Specific comments 2:

55: “continental-ocean” -> “continent-ocean” (twice)

Reply:

We accept the reviewer’s suggestion and have changed “continental-ocean” to “continent-ocean” in Lines 56-57.

Specific comments 3:

165: “fills into” -> “lies within”

Reply:

We accept the reviewer’s suggestion and have changed “fills into” to “lies within” in Line 158.

Specific comments 4:

166: “is different” -> “are different”

Reply:

We accept the reviewer’s suggestion and have changed “is different” to “are different” in Line 159.

Specific comments 5:

168: “has smaller” -> “have smaller”

Reply:

We accept the reviewer’s suggestion and have changed “has smaller” to “have smaller” in Line 161.

Specific comments 6:

171: “dome” -> “doming”

Reply:

We accept the reviewer’s suggestion and have changed “dome” to “doming” in Line 164.

Specific comments 7:

210-211: Rephrase as “The presence of volcanic material has been inferred previously from wide-angle seismic data”.

Reply:

We have rephrased the sentence as “The presence of volcanic material has been inferred previously from wide-angle seismic data” in Lines 296-298.

Specific comments 8:

Figure 2 caption: The supplementary material where the orientation data are given should be cited here.

Reply:

We have cited the supplementary material in the caption of Figure 2.

Specific comments 9:

Fig. 4 caption: "Dimension and aspect ratio" -> "Dimensions and aspect ratios". "S reflector" -> "the S reflector" and later "reflectors" -> "reflector" (twice). The caption should also cite the supplementary material where the origin of the measurements is given.

Reply:

We have changed "Dimension and aspect ratio" to "Dimensions and aspect ratios". We have changed "S reflector" to "the S reflector" and "reflectors" to "reflector". We have also cited the supplementary material in the Fig. 4 caption.